# Cellular and Molecular Differences between HFpEF and HFrEF: A Step Ahead in an Improved Pathological Understanding

**DOI:** 10.3390/cells9010242

**Published:** 2020-01-18

**Authors:** Steven J. Simmonds, Ilona Cuijpers, Stephane Heymans, Elizabeth A. V. Jones

**Affiliations:** 1Center for Molecular and Vascular Biology, KU Leuven, Herestraat 49, bus 911, 3000 Leuven, Belgium; steven.simmonds@kuleuven.be (S.J.S.); ilona.cuijpers@kuleuven.be (I.C.); s.heymans@maastrichtuniversity.nl (S.H.); 2Department of Cardiology, Cardiovascular Research Institute (CARIM), Maastricht University Medical Centre, Universiteitssingel 50, 6229 ER Maastricht, The Netherlands; 3Netherlands Heart Institute, Holland Heart House, Moreelsepark 1, 3511 Utrecht, The Netherlands; 4William Harvey Research Institute, Barts Heart Centre, Queen Mary University of London, Charterhouse Square, London EC1M 6BQ, UK

**Keywords:** heart failure with preserved ejection fraction, heart failure with reduced ejection fraction, inflammation, endothelial dysfunction, cardiomyocyte alterations

## Abstract

Heart failure (HF) is the most rapidly growing cardiovascular health burden worldwide. HF can be classified into three groups based on the percentage of the ejection fraction (EF): heart failure with reduced EF (HFrEF), heart failure with mid-range—also called mildly reduced EF— (HFmrEF), and heart failure with preserved ejection fraction (HFpEF). HFmrEF can progress into either HFrEF or HFpEF, but its phenotype is dominated by coronary artery disease, as in HFrEF. HFrEF and HFpEF present with differences in both the development and progression of the disease secondary to changes at the cellular and molecular level. While recent medical advances have resulted in efficient and specific treatments for HFrEF, these treatments lack efficacy for HFpEF management. These differential response rates, coupled to increasing rates of HF, highlight the significant need to understand the unique pathogenesis of HFrEF and HFpEF. In this review, we summarize the differences in pathological development of HFrEF and HFpEF, focussing on disease-specific aspects of inflammation and endothelial function, cardiomyocyte hypertrophy and death, alterations in the giant spring titin, and fibrosis. We highlight the areas of difference between the two diseases with the aim of guiding research efforts for novel therapeutics in HFrEF and HFpEF.

## 1. Introduction

Heart failure (HF) is the most prominent cause of hospitalization globally, with 3.6 million newly diagnosed patients annually, resulting in a socioeconomic burden of billions of euros per year [1]. Heart failure with preserved ejection fraction (HFpEF) is a complex cardiovascular syndrome presenting with an ejection fraction (EF) of greater than 50%, along with different pro-inflammatory and metabolic co-morbidities. It is characterised by structural and cellular alterations, including cardiomyocyte hypertrophy, fibrosis, and inflammation, all leading to an inability of the left ventricle to relax properly. In contrast, HFrEF, defined by an EF of less than 40%, is characterized by substantial cardiomyocyte loss, resulting in the development of systolic dysfunction; in other words, the inability of the left ventricle to contract properly. Heart failure with mid-range or mildly reduced EF (HFmrEF), is an intermediate stage, with an EF between 40–49%, that generally progresses either to HFpEF (25% of cases) or HFrEF (33% of cases) [2]. With regard to ischaemic aetiology, HFmrEF more resembles HFrEF, but HFmrEF has a higher frequency of underlying coronary artery disease (CAD) and better overall prognosis [2,3,4].

HFpEF is preceded by chronic comorbidities, such as hypertension, type 2 diabetes mellitus (T2DM), obesity, and renal insufficiency, whereas HFrEF is often preceded by the acute or chronic loss of cardiomyocytes due to ischemia, a genetic mutation, myocarditis, or valvular disease [5,6]. This alteration in risk factors already highlights the potential for differing cellular and molecular pathophysiologies of the two diseases. Medical advances have developed efficient and specific treatments of HFrEF by acting on the neuro-humoral axis, but efficacious drugs for the treatment of HFpEF are absent [7]. As a consequence, the prevalence rate of HFrEF has significantly declined over the last few decades, while the prevalence of HFpEF accounts for more than 50% of all HF cases and is expected to rise even further [8]. These differential response rates to therapies in patients with HFrEF and HFpEF underline their distinct underlying cellular and molecular mechanisms [9]. 

## 2. Differences in Comorbidities/Risk Factors in HFrEF and HFpEF

Despite the fact that acute cardiovascular events associated with HFrEF and HFpEF share many risk factors [10], some comorbidities differ between them (Table 1, Figure 1). Patients with HFpEF are more likely to be older [11], with a two-fold predominance of females [12]. This predominance of men in HFrEF might be the result of a greater susceptibility to developing myocardial infarction (MI) [11]. Additionally, men more easily develop eccentric left ventricular hypertrophy upon pressure-overload, while concentric hypertrophy is more common in females [13]. Patients with HFpEF have a higher prevalence of non-cardiac comorbidities (i.e., hypertension, T2DM, stroke, anaemia, pulmonary disease, liver disease, sleep apnoea, gout, and cancer) than HFrEF patients [14]. The mortality risk of the comorbidities studied in both types of HF is similar, regardless of the EF [15,16,17]. Interestingly, in HFpEF, the incidence of hospitalisation for comorbidity-related illness is higher compared to HFrEF [18]. 

## 3. Systemic and Cardiac Inflammation has a Differential Role in HFrEF and HFpEF

Inappropriate or exaggerated inflammation is a key component underlying damage to the heart in both HFrEF and HFpEF, but has different roles in the onset and propagation. Depending on the trigger, the inflammatory response can be classified into three groups: sterile-, metabolic risk-, and non-sterile-induced inflammation (Table 1, Figure 2).

### 3.1. Sterile Inflammation

Sterile inflammation occurs due to post-ischemic or toxic necrosis, massive trauma, haemorrhage, and/or resuscitation, such as in the HFrEF condition MI [76] (Figure 2). During this response, sterile endogenous stimuli trigger inflammation by a) activating the pathogen recognition receptor (PRR) pathway via the exposure to host-derived non-microbial stimuli released following tissue injury, called damage-associated molecular patterns (DAMPs; e.g., DNA, ATP, and hyaluronan), b) releasing intracellular cytokines, which activate common pathways downstream of PRRs, and/or c) triggering the activation of receptors which are not typically associated with microbial recognition, such as the cluster of differentiation 36 (CD36) [19,77]. Following ligand recognition, PRRs activate downstream signalling, such as transcription factor nuclear factor kappa-B (NFκB), mitogen-activated kinase (MAPK), and type I interferon (IFN), which upregulate pro-inflammatory cytokines and chemokines that are important in the healing response [19]. Furthermore, dying cells release molecules that break the extracellular matrix (ECM) down (matrix metalloproteinases; MMPs), oxidize macromolecules, and induce the production and release of abnormal host cell products, such as heat shock proteins (HSPs) [76,78]. 

In the meantime, the activation of adrenergic signalling alerts the bone marrow to reduce the production of hematopoietic stem cell (HSC) retention factors [79]. Consequently, HSC and progenitor cells are attracted from their specific niches in the bone marrow by the stromal cell-derived chemokine CXCL12, leading to the production of neutrophils, called granulopoiesis [79]. Neutrophils are rapidly released in the blood and are recruited to the heart, where they phagocytose the damaged cells. Interestingly, the serum neutrophil to lymphocyte ratio is considered a useful marker for the prediction of mortality in patients admitted for ST-segment elevation MI (STEMI) [80]. Furthermore, HSCs egress from the bone marrow and seed in the spleen, thereby triggering extramedullary hemopoiesis and monocyte production [81]. Within 24 h after MI, the splenic monocyte reservoir is released into the circulation and monocytes are recruited to the heart [82]. Monocytes subsequently infiltrate and produce cytokines and growth factors, such as IL-10 and transforming growth factor β (TGFβ), to repress inflammatory signalling, promote new blood vessel formation by endothelial cells and vascular smooth muscle cells, and scar formation by (myo)fibroblast activation [83,84]. Furthermore, B- and total and regulatory T-cells are recruited to the heart, where they modulate cardiac remodelling [19,85,86]. Regulatory T-lymphocytes have a protective role in ventricular remodelling in MI [86]. As such, the adoptive transfer of regulatory T-cells attenuates inflammation by limiting the local inflammatory response, including reducing immune cell infiltration (T-lymphocytes, macrophages and neutrophils) and pro-inflammatory cytokine production by cardiomyocytes [86]. However, the complex interplay between injured cardiomyocytes and the innate and adaptive immune response is often insufficient to prevent damage to the cardiac contractility.

### 3.2. Metabolic Risk-Induced Inflammation

While the inflammatory response in HFrEF is the result of cardiomyocyte damage, rather than the systemic comorbidities, inflammation in HFpEF is the result of a cluster of extra-cardiac metabolic and inflammatory risk factors, including obesity, diabetes, anaemia, hypertension, chronic obstructive pulmonary disease (COPD), auto-immune diseases, and renal insufficiency [9] (Figure 2). Obese and type 1 and 2 diabetic patients have, for example, elevated levels of advanced glycation end products (AGEs) [87,88,89]. These AGEs interact with their receptor (RAGE), thereby activating the NFκB signalling pathway and secondary pro-inflammatory cytokines, chemokines, and adhesion molecules [90]. 

The “systemic microvascular paradigm” for HFpEF development proposes that low-grade chronic systemic inflammation initiates detrimental microvascular changes [9]. In line with this, circulating protein levels of pro- inflammatory markers interleukin-6 (IL-6), tumour necrosis factor alpha (TNF-α), and circulating acute inflammatory C-reactive protein (CRP) were higher and more predictive in HFpEF compared to HFrEF [44,91]. These CRP levels were also proportional to the number of HFpEF-associated comorbidities [92]. Importantly, HFpEF patients also had elevated circulating levels of neutrophils and classical, intermediate, and non-classical monocytes, while circulating lymphocytes were not affected [93,94]. Correspondingly, the culturing of healthy donor monocytes with serum from patients with HFpEF promoted the differentiation towards alternative activated pro-fibrotic macrophages [94]. In patients with HFpEF, echocardiographic indices of diastolic dysfunction (E/e’) correlated with monocyte recruitment in the spleen [93]. This monocyte recruitment to the heart may result in part by increased expression of endothelial adhesion molecules, such as E-selectin and intercellular adhesion molecule 1 (ICAM-1). Their expression is induced directly by comorbidities and indirectly by chronic low-grade inflammation [25,90,93,95,96,97,98,99,100,101,102]. 

### 3.3. Non-sterile inflammation 

Non-sterile inflammation can be induced by a viral infection. Viral myocarditis (VM), for example, is characterized by a cardiac infection caused by a normally benign virus, triggering an exaggerated immune response in immune-susceptible persons. The initial acute phase is characterized by increased circulating and cardiac neutrophils and monocytes, while in later phases, increased circulating T-cells, specifically cytotoxic T-cells, and B-cells can be detected [20,21]. This results in scar tissue formation and a reduced contractile function in the chronic phase [21]. VM may result in both systolic and diastolic dysfunction in the long term. 

In short, regardless of the cause of HFrEF, inflammation results in cardiac remodelling driven by cardiomyocyte damage or loss, which causes cardiomyocyte autophagy, apoptosis, and necrosis [9,28,29,30]. In contrast, metabolic risk factors in HFpEF drive the chronic low-grade systemic inflammation along with the increased expression of adhesion molecules on the endothelial cells, leading to systemic and local cardiac inflammation

## 4. Endothelial Dysfunction: a Main Player in Both HFrEF and HFpEF 

Endothelial cells represent the majority of non-cardiomyocyte cells (60%). Structural and/or functional abnormalities will therefore strongly impact cardiac function [103]. Endothelial dysfunction is an early event in the development of cardiovascular diseases, such as HF, and is therefore an important indicator of survival [104,105]. It is present in both HFpEF and HFrEF (Table 1) [23,24,106], as shown by a similar loss in cardiac microvascular density [23,24]. However, comorbidities can differentially affect endothelial function, with microvascular complications being more prevalent in HFpEF than in HFrEF [22]. Besides being more severe in HFpEF, endothelial dysfunction happens earlier in the progression of HFpEF. It is therefore an early marker in HFpEF, while being a late stage symptom in HFrEF [9]. Therefore, future research should focus on the early stages of endothelial dysfunction in HFpEF, before the full disease has developed. 

Increased oxidative stress, imbalanced nitric oxide (NO) bioavailability, or neurohormonal activation and vasoconstriction in response to reduced cardiac output (CO) are central mechanisms causing endothelial dysfunction [107,108,109,110,111,112,113]. Imbalanced NO bioavailability can result from oxidation of NO and reduced NO production, due to, for example, the uncoupling of endothelial nitric oxide synthase (eNOS), all caused by oxidative stress [9]. HFpEF myocardial biopsies showed reduced NO bioavailability and increased eNOS uncoupling compared to HFrEF patients [25]. Imbalanced NO bioavailability and oxidative stress (nitroso-redox imbalance) subsequently reduce the coronary endothelium-dependent vasodilator capacity [113,114,115], resulting in impaired myocardial perfusion and coronary flow [113]. Interestingly, in patients with chronic HFrEF, endothelium-mediated vasodilation is an independent predictor of cardiac death and hospitalization, consistent with the notion that endothelium-derived NO might have a protective role in chronic HFrEF [107]. In HFpEF, despite the microvascular paradigm, endothelial dysfunction is not sufficiently well evaluated [9]. Reduced NO signalling is proposed to influence adjacent cardiomyocytes, leading to reduced cyclic guanosine monophosphate (cGMP) and protein kinase G activity (PKG) and increased protein phosphatases 1 and 2a (PP1 and 2a, respectively) activity [25]. This altered signalling is thought to modulate cardiomyocyte hypertrophy and stiffness in HFpEF [9]. 

In addition to an impaired NO bioavailability, there is an association between endothelial dysfunction and reduced endogenous production of estrogens after menopause [116]. Endogenous estrogens have direct and indirect effects on the cardiovascular system. Estrogens directly modulate the vascular tone by a) acute vasodilation by increasing the synthesis of NO [117], b) long-term modulation by regulation of the production of prostaglandins and expression of eNOS and endothelin-1 (ET-1) [118,119], c) inhibition of endothelin-induced vasoconstriction [120], and d) inhibition of sympathetic activity [121]. Furthermore, estrogens have indirect effects on the cardiovascular system by the modulation of cardiovascular risk factors, as women have shown increased total and low-density lipoprotein (LDL) within one year after their final menstrual period [122]. Furthermore, estrogen has anti-proliferative effects on the vascular smooth muscle layer [123], modulates vascular remodelling [124], and exerts anti-oxidative effects [125]. In men with stable HFrEF, both low and high concentrations of circulating estrogen were significant predictors of a poor prognosis [126]. In murine ischemic/reperfusion, estrogen improved functional recovery, decreased cardiomyocyte apoptosis, and necrosis [127]. Notably, HFpEF patients are mostly elderly post-menopausal women, suggesting that a loss of estrogen could partly mediate cardiovascular effects in HFpEF. Interestingly, estradiol administration significantly improved left ventricular diastolic function in hypertensive postmenopausal women with left ventricular diastolic dysfunction [128]. Furthermore, long term exposure to (phyto)estrogen protected against murine HFpEF development [129]. However, the exact mechanism of the lack of estrogen in HFpEF development is unclear. 

## 5. Cardiomyocyte Alterations in HFrEF and HFpEF: Eccentric Versus Concentric Hypertrophy

During HFrEF, remodelling is driven by cardiomyocyte damage and loss, leading to an imbalance in the heart wall structure, causing eccentric remodelling, with left ventricular dilation, but a normal wall thickness (Table 1) [26,27,28,29,30]. In contrast, HFpEF patients reveal concentric cardiomyocyte hypertrophy, where the left ventricle wall thickens (Table 1) [31,32]. Remodelling differences in HFrEF and HFpEF can be explained by structural differences in the cardiomyocytes, with thinner and more elongated cardiomyocytes, lower myofibrillar density, and reduced cardiomyocyte passive stiffness (F_passive_) in cardiac biopsies of HFrEF patients compared to HFpEF patients [54]. Comorbidities found in both HFrEF and HFpEF also differentially alter the structure of the heart; T2DM, for example, causes a more pronounced left ventricular hypertrophy and worsened quality of life in HFpEF compared to HFrEF [22]. Furthermore, increased cardiomyocyte size in the HFpEF biopsies correlates with lower myocardial PKG activity and cGMP concentration [130]. In contrast, circulatory levels of insulin-like growth factor 1 (IGF-1), a vasoprotective hormone, which stimulates myocardial contractility, were reduced in HFrEF compared to HFpEF [131], suggesting a reduced protection of cardiomyocytes from oxidative and hypertrophic stress [9,132]. Thus, differences in stimuli and altered cellular signalling contribute to the distinct types of cardiac hypertrophy observed in HFrEF and HFpEF. 

## 6. Cardiomyocyte Cell Death: a Typical Characteristic of HFrEF

Cardiomyocyte damage in HFrEF patients, indicated by increased circulating levels of Troponin-T, leads to a reduction in functioning cardiomyocyte mass in combination with excessive fibrotic tissue [9,33,34,35,36]. Cardiomyocyte loss during HFrEF occurs by apoptosis, necrosis, necroptosis, or autophagy depending on the underlying cause [133]. Ischemia induces both apoptosis and necrosis [134]. In VM, an inappropriate inflammatory response causes cardiomyocyte loss [20]. Pressure-overload may also induce cardiomyocyte apoptosis, in part by the pro-apoptotic FAS (also known as apoptosis antigen 1), its ligand, caspase-8, and its cleavage enzyme [34,135]. Additionally, in a type 1 diabetes mellitus (T1DM)-induced model of diabetic cardiomyopathy, mitochondrial regulatory proteins were upregulated during HFrEF, suggesting mitochondrial dysregulation, which could lead to further cardiomyocyte apoptosis [136]. Cardiac steatosis is reported in both HFrEF and HFpEF patients and models, and is also hypothesized to cause cardiomyocyte apoptosis [137,138]. Lipid exposure has been shown to cause cardiomyocyte apoptosis in vitro [139]; however, current studies show conflicting evidence in vivo [140,141,142]. Recent evidence has suggested that cardiomyocyte autophagy and necroptosis (a controlled form of necrosis) play a larger role than apoptosis [143]. Interestingly, even low levels of cardiomyocyte apoptosis induce HFrEF in animal models [144]. In contrast, circulating Troponin-T, which indicates elevated levels of cardiomyocyte cell death, was not detected in HFpEF patients [37]. However, in a rat model of HFpEF, the inhibition of apoptotic and autophagic genes improved diastolic dysfunction, even though there was no evidence of decreased apoptosis or autophagy [145]. Thus, though cardiomyocyte cell death is a characteristic observed in HFrEF and not in HFpEF (Table 1), these data suggest that apoptosis and autophagy pathways may also play a role in cardiomyocyte hypertrophy and stiffening during HFpEF, without leading to cardiomyocyte death. 

## 7. Different Types of Cardiac Fibrosis in HFrEF and HFpEF 

In the healthy heart, the ECM is mainly composed of thicker collagen type I fibres (~85%), which confer tensile strength, and thin collagen type III fibres (~11%), that maintain the elasticity of the matrix network [146]. In addition to collagens, the cardiac ECM also contains glycosaminoglycans (e.g., hyaluronan), glycoproteins (e.g., SPARC, thrombospondin, fibrillin and fibronectin), and proteoglycans (e.g., syndecan, osteoglycin). Under pathophysiological stresses, such as aging, hypertension, T2DM, myocardial injury, toxic insults, and pressure overload, augmented cardiac fibrosis develops (Table 1). In HFrEF, cardiac fibrosis is mainly the consequence of cardiomyocyte loss and may contribute to systolic dysfunction via several mechanisms. First, the loss of fibrillar collagen may impair the transduction of cardiomyocyte contraction into myocardial force, resulting in the uncoordinated contraction of cardiomyocyte bundles [147]. Second, cardiac fibrosis impairs interactions between endomysial components such as laminin, which connect adjacent cardiomyocytes and capillaries, and their receptors, thereby disrupting the cardiomyocyte mass and systolic reserve [148]. This contrasts with HFpEF, where excessive collagen deposition and a reduction of the more flexible collagen III results in increased cardiac stiffness [40,149]. 

Activated myofibroblasts, important players in the fibrotic processes, are increased in the myocardium during HFrEF (e.g., MI and pressure-overload) and HFpEF-associated comorbidities (e.g., T2DM, and aging) [150,151,152,153]. Despite showing myofibroblast accumulation, the signals which control fibrosis are different. In the healthy heart, TGFβ1, a central mediator in fibrogenesis, is present as a latent complex, but is activated during cardiac injury by a wide range of mediators, such as proteases (e.g., plasmin, MMP-9), thrombospondin-1 (TSP-1), integrins, and ROS [154,155,156,157]. Interestingly, the ratios of plasma levels of MMP-9 to tissue inhibitor of MMP-1 (TIMP-1) were higher in HFrEF compared to HFpEF patients, while no differences were found in MMP-9 levels in HFrEF versus HFpEF [158]. This suggests that this differential MMP/TIMP signaling in HFrEF and HFpEF might result in distinctive TGF β1 signalling. TGFβ1 signalling subsequently activates genes responsible for myofibroblast transdifferentiation and ECM synthesis, thereby increasing fibrosis. Patients with HFpEF exhibited higher serum concentrations of the pro-fibrotic TGFβ1 protein compared to HFrEF patients [43]. In addition, chemokines, such as monocyte chemoattractant protein 1 (MCP-1/CLL-2) recruit pro-fibrotic leukocyte subpopulations, fibroblasts progenitors, and control fibroblast behaviour [159,160,161,162]. HFpEF patients have increased MCP-1/CLL-2 serum levels, which are reduced in HFrEF [44,45,46]. 

The effects of cardiac fibrosis are not solely due to amount of collagen deposited, but also the location. Perivascular fibrosis develops as a response to the production of pro-fibrotic proteins (e.g., TGFβ) by infiltrated immune cells close to the vasculature [9]. This fibrosis is the consequence of a systemic and cardiac inflammation in both HFpEF and HFrEF, but is significantly greater in HFpEF [38]. Interstitial fibrosis is characterized by the expansion of the ECM surrounding the cardiomyocytes, occurring in HF without cardiomyocyte loss [163], including non-ischemic HFrEF and HFpEF [39,40,41]. Unlike other organs, the heart has a very limited regenerative capacity, therefore repair processes involving the removal of necrotic cardiomyocytes followed by replacement fibrosis aim to preserve myocardial structural and functional integrity [164]. In the early inflammatory phase after MI, a temporary fibrin matrix (granulation tissue) is formed to replace dead cells [42]. In the subsequent phases, this is replaced by a collagen-based matrix, leading to the development of a mature scar, which is mostly devoid of cardiomyocytes. This mature scar consists of a mature dense collagen network (mainly collagen I and III) containing fibroblasts, immune cells, and microvessels [42,165]. Cardiomyocyte loss is predominant in HFrEF, and not reported in HFpEF, as described before, therefore this type of replacement fibrosis is only present in HFrEF. 

Collagen crosslinking is also relevant to fibrotic disease pathophysiology, as it is differentially regulated between HFrEF and HFpEF. HFpEF patients show increased collagen cross-linking and upregulation of the collagen crosslinker lysyl-oxidase (LOX), associated with impaired diastolic tissue Doppler parameters (e.g., E/E’) [40]. Currently, no studies have measured levels of LOX in HFrEF; however, trials using crosslink breakers showed a decrease in left ventricular stiffness in HFpEF patients, but no efficacy, and even increased cardiac dilatation in HFrEF patients [166,167]. Secreted ECM glycoproteins and proteoglycans also modulate collagen properties. Plasma syndecan-1 levels correlate with cardiac fibrosis in patients with HFpEF and HFrEF and were independently associated with clinical outcomes in patients with HFpEF but not in patients with HFrEF [168]. Furthermore, SPARC is involved in post-synthetic procollagen processing and the formation of mature cross-linked collagen fibrils in pressure-overloaded and aged hearts [169,170]. Lastly, osteoglycin negatively regulates cardiac fibrotic remodelling by attenuating cardiac myofibroblast proliferation and migration [171]. Thus, differences in the localization, composition, and crosslinking of the cardiac fibrous tissue contribute to the differences in HFrEF and HFpEF. 

## 8. Left Ventricular Stiffness in HFrEF and HFpEF: Cardiac Titin and Ca^2+^ Levels are Differently Affected in HFpEF and HFrEF

Increased left ventricular stiffness leading to diastolic dysfunction is a hallmark of HFpEF [66]. However, HFrEF risk factors also alter left ventricular stiffness (Table 1). For example, patients with CAD or MI had an impaired active relaxation [50,172]. In rats, chronic coronary occlusion initially prolonged relaxation, which returned to normal after one hour of occlusion until five days after MI and then increased permanently, similarly to the HFpEF phenotype [173]. In patients with MI, left ventricular stiffness may vary, going from normal, to increased, or even reduced levels [47,48,49,50,51,52]. Patients with end-stage non-ischemic dilated cardiomyopathy showed reduced left ventricular stiffness [53]. Changes in ventricular stiffness arise not only from cardiac fibrosis (discussed above), but also from cardiomyocyte F_passive_ [66,174,175]. Cardiomyocytes from HFpEF patients have a significant increase in F_passive_, when compared to those from HFrEF patients [54,55]. Changes in cardiomyocyte F_passive_ occur through several mechanisms which are alternatively regulated between HFrEF and HFpEF.

### 8.1. Titin Modifications in HFrEF and HFpEF

Titin, a bidirectional giant spring, is the main determinant of cardiomyocyte F_passive_. Alternative splicing of titin mRNA generates two isoforms: the shorter but stiffer N2B, and the longer more flexible N2BA (Figure 3A). Animal models for HFpEF, obesity, T1DM, T2DM, and hypertension, showed isoform switching from the more flexible N2BA to the stiffer N2B, which is associated with an increased F_passive_ [67,68,69,70,71]. In contrast, in cardiac biopsies from HFrEF patients, the flexible N2BA isoform was increased, while total titin levels were unchanged, indicating a switch from the stiffer N2B to the more flexible N2BA isoform and a reduced F_passive_ [56]. 

In contrast to isoform shifts, post-translation modifications of titin provide a more rapid mechanism to affect cardiomyocyte’s F_passive_ (Figure 3B). Hypophosphorylation of total N2B isoform increased cardiomyocyte F_passive_ in human HFpEF biopsies and animal models for HFpEF [69,72,73]. The in vitro administration of protein kinase A (PKA) decreased F_passive_ in human cardiomyocytes by increasing the phosphorylation of N2B [176]. Furthermore, site-specific phosphorylation of titin in different regions, including the N2B unique sequence (N2Bus) and a region rich in proline, glutamate, valine, and lysine (PEVK), affects cardiomyocyte F_passive_ [177]. In patients with HFpEF and hypertension, titin was hypophosphorylated at S4185 (PKG/PKA site in N2Bus), but hyperphosphorylated at S11878 (PKCα site in PEVK) compared to non-hypertensive HFpEF patients and control subjects [74]. In patients with HFrEF, titin dephosphorylation and aggregation ex vivo were also partially responsible for increased F_passive_, which could increase overall cardiac stiffness [57]. Interestingly, truncated titin variants in HFrEF patients promoted cardiac functional and structural alterations [58]. In myocardial ischemia/reperfusion injury, the protease MMP-2 cleaved titin [59], suggesting a role of proteolysis in decreasing cardiomyocyte passive stiffness during HFrEF. 

### 8.2. Cardiac Ca^2+^ Levels in HFpEF and HFrEF

Cardiac relaxation also depends on a decreased level of intracellular calcium (Ca^2+^). Sequestration of cytoplasmic Ca^2+^ mainly occurs through active Ca^2+^ uptake by the sarcoplasmic reticulum (SR) via Ca^2+^ ATPase 2a (SERCA2a). Extrusion to the extracellular space via Na^+^/Ca^2+^ exchanger (NCX), sacrolemmal SERCA, and mitochondria are also involved [178,179,180,181]. In HFrEF, impaired Ca^2+^ release from the SR leads to systolic dysfunction, as cardiac contraction depends on a transient increase in [Ca^2+^]_I_ to activate cross-bridge formation between myofilament proteins. For example, patients with end-stage HFrEF have elevated myocardial [Na^+^]_I_ [60]. Small increases in [Na^+^]_I_ may increase the Ca^2+^ influx via reverse mode NCX and limit the Ca^2+^ extrusion via forward mode NCX, leading to elevated levels of sub-sarcolemmal [Ca^2+^]_I_ [182]. Furthermore, both reduced SERCA2a activity and inhibition of SERCA2 by phospholamban are observed in HFrEF, leading to reduced Ca^2+^ removal [61,62,63,64,65]. Myocardial Ca^2+^ levels are increased in patients with HFpEF, indicating reduced Ca^2+^ removal, as observed in HFrEF. However, unlike HFrEF, this deficit in Ca^2+^ removal in HFpEF is not dependent on the Na^+^ gradient [75]. These results highlight how slight changes in ion handling have a profound effect on cardiac function. 

## 9. Conclusions

Despite the fact that HFrEF and HFpEF share many risk factors and co-morbidities, there are substantial differences in systemic inflammation, cardiac remodelling (endothelial function, hypertrophy, and cardiomyocyte cell death), and stiffness (fibrosis, titin, and calcium levels). HFrEF, mostly occurs in male patients and is the consequence of cardiomyocyte loss. In contrast, HFpEF is often diagnosed in elderly female patients, who suffer from (a cluster of) non-cardiac comorbidities, such as hypertension, T2DM, stroke, anaemia, pulmonary disease, liver disease, sleep apnoea, gout, and cancer. HFpEF is characterized by low-grade chronic systemic inflammation and capillary dysfunction, with consequential low-grade cardiac inflammation. However, as most studies are conducted on models of established HFpEF, studies during the disease onset are required to elucidate the common denominator underlying the associated complications for HFpEF. In HFrEF, systemic and cardiac inflammation are secondary to the causes of cardiomyocyte loss. Whereas in HFpEF endothelial dysfunction mainly precedes its progression, in HFrEF endothelial dysfunction may rather be the consequence. Depending on the stimulus, perivascular (metabolic risks in HFpEF), interstitial (e.g., aging, hypertension in both HFpEF and HFrEF), or replacement (e.g., MI in HFrEF) fibrosis occurs and is differentially managed between the two conditions. The increased left ventricular stiffness in HFpEF is caused by reduced Ca^2+^ signalling, titin modifications (isoform shifts towards the stiffer isoform and post-translational changes), and increased perivascular and interstitial fibrosis. In contrast, in HFrEF, titin isoform switching is less consistent, and more flexible isoforms even present in end-stage non-ischemic dilated cardiomyopathy, therefore left ventricular stiffness ranges from being unaffected, to increased, or even reduced.

## Figures and Tables

**Figure 1 cells-09-00242-f001:**
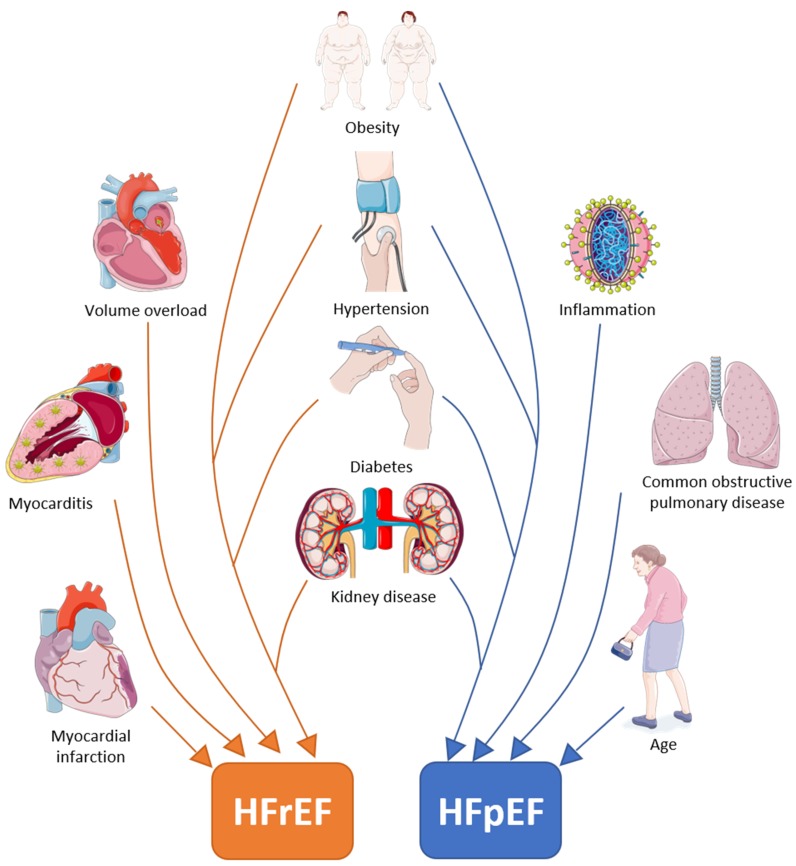
Risk factors and comorbidities involved in the development of either heart failure with reduced ejection fraction, heart failure with preserved ejection fraction or both. Image created using artwork from Servier medical art.

**Figure 2 cells-09-00242-f002:**
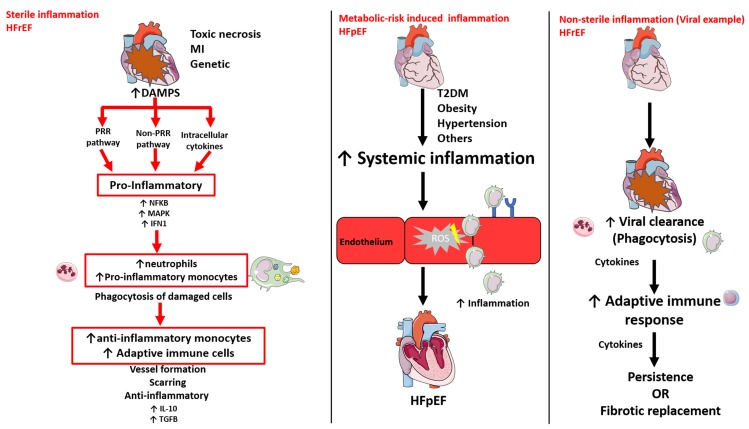
Schematic of sterile-, metabolic risk-, and non-sterile-induced inflammation in the development of heart failure with reduced ejection fraction and heart failure with preserved ejection fraction. HFpEF (heart failure with preserved ejection fraction), HFrEF (heart failure with reduced ejection fraction), HSC (haemopoietic stem cell), IFN1 (interferon 1), IL-10 (interleukin 10), MAPK (mitogen-activated protein kinase), NFκB (nuclear factor kappa B), PRR (pathogen recognition receptor), T2DM (type 2 diabetes mellitus), ROS (reactive oxygen species), TGFβ (transforming growth factor beta) Image created using artwork from Servier medical art.

**Figure 3 cells-09-00242-f003:**
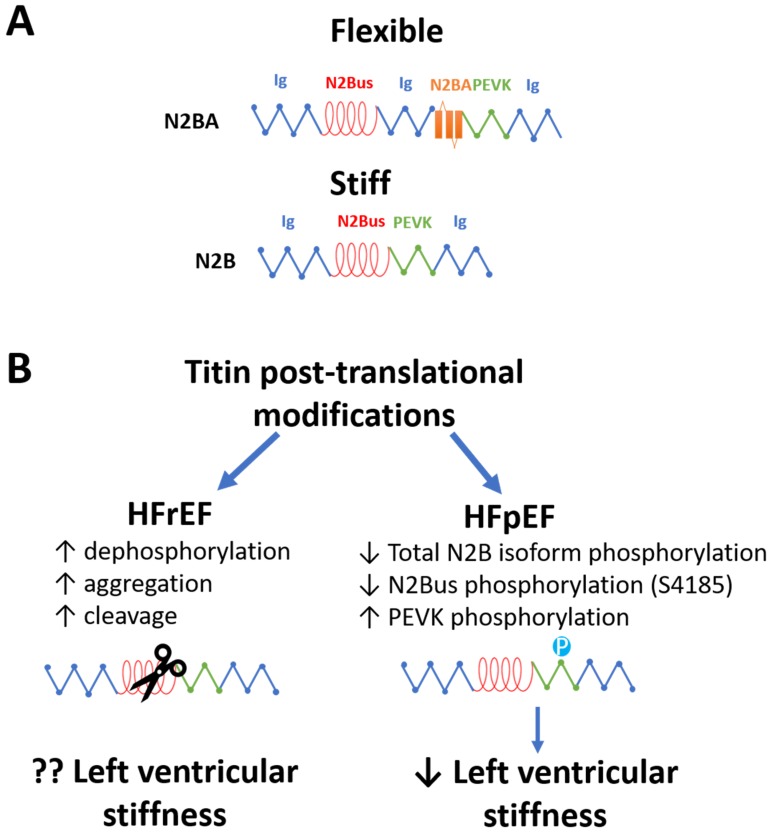
The role of titin in left ventricular stiffness. (**A**) Diagram of the alternative isoforms of titin. (**B**) Post-translational modifications of Titin and their effect on left ventricular stiffness in HFrEF and HFpEF.

**Table 1 cells-09-00242-t001:** Summary of physiological, cellular and molecular similarities and differences between HFrEF and HFpEF.

	HFrEF	HFpEF
**Comorbidities/risk factors**	Obesity Hypertension Diabetes Kidney disease Male Volume overload Myocarditis Myocardial infarction [11]	Obesity Hypertension Diabetes Kidney disease Female COPD Age Anaemia Inflammation Liver disease Sleep apnoea Gout Cancer [10,11,12,14]
**Systemic and cardiac inflammation**	Sterile [19] Non-sterile [20,21]	Metabolic-risk induced [9]
**Endothelial dysfunction**	Late stage symptom [9] ↓ Prevalence [22] ↓ Microvascular density [23,24] ↓ NO bioavailability [25]	Early stage symptom [9] ↑ Prevalence [22] ↓ Microvascular density [23,24] ↓↓ NO bioavailability [25]
**Cardiac hypertrophy**	Eccentric [26,27,28,29,30]	Concentric [31,32]
**Cardiomyocyte cell death**	Present [9,33,34,35,36]	Absent [37]
**Cardiac fibrosis**	↑ Perivascular fibrosis [38] Interstitial fibrosis [39,40,41] Replacement fibrosis [42] Collagen crosslinking? *Fibrotic signalling:* ↑ Serum TGFβ1 [43] ↓ MCP-1 [44,45,46]	↑↑ Perivascular fibrosis [38] Interstitial fibrosis [39,40,41] ↑ Collagen crosslinking [40] *Fibrotic signalling:* ↑↑ Serum TGFβ1 [43] ↑ MCP-1 [44,45,46]
**Left ventricular stiffness**	Left ventricular stiffness? [47,48,49,50,51,52,53] *Titin:* Cardiomyocyte F_passive_? [54,55] ↑ N2BA [56] Titin dephosphorylation [57] Titin aggregation [57] Titin cleavage [58,59] ***Cardiac calcium signalling:*** ↑ Myocardial [Na^+^]_I_ [60] ↓ SERCA2a activity [61,62,63,64,65]	↑ Left ventricular stiffness [66] *Titin:* ↑ Cardiomyocyte F_passive_ [54,55] N2BA → N2B [67,68,69,70,71] N2B hypophosphorylation [69,72,73] N2Bus hyperphosphorylation at S4185 [74] PEVK hyperphosphorylation at S11878 [74] ***Cardiac calcium signalling:*** ↑ Myocardial Ca^2+^ levels [75]

Ca^2+^ (calcium ion), COPD (chronic obstructive pulmonary disease), Fpassive (passive tension), MCP-1 (monocyte chemoattractant protein 1), Na^+^ (sodium ion), NO (nitric oxide), SERCA2a (sarcoplasmic reticulum via Ca^2+^ ATPase 2a), TGFβ1 (Transforming growth factor beta 1).

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
