# Peer review of "Cellular and Molecular Differences between HFpEF and HFrEF: A Step Ahead in an Improved Pathological Understanding"

_cells, 2020, doi:10.3390/cells9010242_

Round 1
Reviewer 1 Report
Authors reviewed the tissue and cell mechanisms involved in HFrEF and HFpEF, analysing several cardiac responses to the related insults. They included interesting data by comparison of their associated mediators. The schemes are very descriptive and summarize the well-designed and written text. Some appreciations could, however, improve the comprehension of the work.
First, metalloproteinases and their inhibitors (TIMPs) may play a significant role in each pathology, being determinant in the fibrotic component. Thus, related data should be included for both HF A final scheme of common and different cardiac molecular mechanisms on inflammation, hypertrophy, apoptosis/necrosis and fibrosis for both HF, should be added. Also, steatosis response may influence cardiac fate in these patients, overall after DM2 and obesity, and even HTN. In this sense, the linked mitochondrial injury can be definitive to initiate apoptosis in cardiac cells. This issue should be mentioned in the manuscript. Suggested references: Fuentes-Antras Card Diabetol 2015, Aon Front Physiol 2014 Could you please hypothesize in the Discussion section what is the common denominator of the associated complications for HFpEF?Author Response
Point 1:
First, metalloproteinases and their inhibitors (TIMPs) may play a significant role in each pathology, being determinant in the fibrotic component. Thus, related data should be included for both HF.
We agree with the reviewer and we have added this to the section “Different types of cardiac fibrosis in HFrEF and HFpEF” (lines 359-362). New abbreviations are added to the list of abbreviations.
Point 2:
A final scheme of common and different cardiac molecular mechanisms on inflammation, hypertrophy, apoptosis/necrosis and fibrosis for both HF, should be added.
We agree with the reviewer. We added a table containing common and different molecular and cellular mechanisms on comorbidities/risk factors, systemic and cardiac inflammation, endothelial dysfunction, cardiac hypertrophy, cardiomyocyte cell death, cardiac fibrosis, and left ventricular stiffness at page (8). The table is referred in the related sections.
Point 3:
Also, steatosis response may influence cardiac fate in these patients, overall after DM2 and obesity, and even HTN. In this sense, the linked mitochondrial injury can be definitive to initiate apoptosis in cardiac cells. This issue should be mentioned in the manuscript. Suggested references: Fuentes-Antras Card Diabetol 2015, Aon Front Physiol 2014.
We have added this to the section entitled “Cardiomyocyte cell death: a typical characteristic of HFrEF” (lines 315-328).
Point 4:
Could you please hypothesize in the Discussion section what is the common denominator of the associated complications for HFpEF?
Thank you for your comment. Currently, most studies are conducted on animal models and human biopsies of established HFpEF. However, studies during the onset and development of HFpEF are required to elucidate the common denominator underlying the associated complications for HFpEF. We have added this to our conclusion in line (473-475).
Reviewer 2 Report
This manuscript by Simmonds et al is different from another review by Chen et al “Heart Failure with Reduced Ejection Fraction (HFrEF) and Preserved Ejection Fraction (HFpEF): The Diagnostic Value of Circulating MicroRNAs” (Cells, 2019, 8:12), and nicely summarized the differences in pathological development of HFrEF and HFpEF, focusing on disease-specific aspects of inflammation and endothelial function, cardiomyocyte hypertrophy and death, alterations in the giant spring titin, and fibrosis. Overall, this review is very comprehensive and well written. There are several minor issues on editing/formatting.
Line 28, “focussing” should be “focusing”. Page 2: more space is remained. Can the text be moved up, and remain figure next page? Or make Figure 1 smaller? Can Figure 2 together with figure legend be shrunk to one page? Lines 189, 225, 255, 290, have more line spaces. Page 5, Can the text be moved up, and remain Figure 3 next page? Lines 360-382: Some sentences use the present tense, and some sentences use the past tense. Please correct. Also, please re-edit this manuscript to eliminate typos. It would be better to mention the review article by Chen et al (Cells, 2019, 8:12).
Author Response
Point 1:
Line 28, “focussing” should be “focusing”.
Thank you for your feedback. We used British English, as such we used the word focussing.
Point 2:
Page 2: more space is remained. Can the text be moved up, and remain figure next page? Or make Figure 1 smaller?
Thank you for your formatting feedback. We did not change this, as we would like to cohere to the formatting style of Cells.
Point 3:
Can Figure 2 together with figure legend be shrunk to one page?
We have diminished the size of figure 2, such that the figure and the legend fit on one page.
Point 4:
Lines 189, 225, 255, 290, have more line spaces.
Thank you for your comment. We have removed the line spacing at line 189. We have not removed the other line spacings, as they are the line spacing between each new section.
Point 5:
Page 5, Can the text be moved up, and remain Figure 3 next page?
Thank you. This is formatted by cells. We would like to be coherent to their style guidelines.
Point 6:
Lines 360-382: Some sentences use the present tense, and some sentences use the past tense. Please correct.
Thank you for your comment. We have corrected this in lines 409-413.
Point 7:
Also, please re-edit this manuscript to eliminate typos.
Thank you. We have reread the manuscript to eliminate typos.
Point 8:
It would be better to mention the review article by Chen et al (Cells, 2019, 8:12).
We read the interesting review article by Chen et al, but could not fit it in our review.